# Role of Biomarkers in the Diagnosis and Treatment of Inflammatory Bowel Disease

**DOI:** 10.3390/life11121375

**Published:** 2021-12-10

**Authors:** Kohei Wagatsuma, Yoshihiro Yokoyama, Hiroshi Nakase

**Affiliations:** Department of Gastroenterology and Hepatology, Sapporo Medical University School of Medicine, Minami 1-jo Nishi 16-chome, Chuo-ku, Sapporo, Hokkaido 060-8543, Japan; yoshi_yokoyamaa@yahoo.co.jp (Y.Y.); hiropynakase@gmail.com (H.N.)

**Keywords:** Crohn’s disease, C-reactive protein, fecal calprotectin, fecal immunochemical test, inflammatory bowel disease, leucine-rich glycoprotein, mucosal healing, prostaglandin E-major urinary metabolite, ulcerative colitis

## Abstract

The number of patients with inflammatory bowel disease (IBD) is increasing worldwide. Endoscopy is the gold standard to assess the condition of IBD. The problem with this procedure is that the burden and cost on the patient are high. Therefore, the identification of a reliable biomarker to replace endoscopy is desired. Biomarkers are used in various situations such as diagnosis of IBD, evaluation of disease activity, prediction of therapeutic effect, and prediction of relapse. C-reactive protein and fecal calprotectin have a lot of evidence as objective biomarkers of disease activity in IBD. The usefulness of the fecal immunochemical test, serum leucine-rich glycoprotein, and urinary prostaglandin E major metabolite have also been reported. Herein, we comprehensively review the usefulness and limitations of biomarkers that can be used in daily clinical practice regarding IBD. To date, no biomarker is sufficiently accurate to replace endoscopy; however, it is important to understand the characteristics of each biomarker and use the appropriate biomarker at the right time in daily clinical practice.

## 1. Introduction

Inflammatory bowel disease (IBD) is a refractory and recurrent inflammatory disease that mainly affects young people, and includes Crohn’s disease (CD) and ulcerative colitis (UC). The number of patients with these diseases is increasing worldwide [1,2]. The initial goal of IBD treatment was to achieve clinical remission; however, this has changed due to advances in methods for assessing IBD activity and treatments. Particularly, recent studies have shown that the goal of IBD treatment is to achieve mucosal healing (MH), which reduces the rate of hospitalization and bowel resection and improves the prognosis of IBD [3,4].

Endoscopic healing is a therapeutic goal in recent clinical trials and clinical practice [5]. Bouguen et al. frequently used endoscopic assessments and coordinated medical treatment as needed to achieve better outcomes for patients with CD [6]. Decreased colonoscopy monitoring rates in patients with IBD increase the risk of disease-related complications [7]. Patients with IBD have reported greater embarrassment and burden of bowel cleansing for colonoscopy and increased pain during colonoscopy [8,9,10]. Fewer colonoscopies can also prevent infections such as coronavirus disease 2019 from being transmitted during endoscopy [11]. Therefore, it is necessary to identify reliable, non-invasive surrogate biomarkers to reduce patient burden and cost.

A biomarker is a biological observation that predicts a clinically relevant endpoint or intermediate outcome in place of an outcome that is more difficult to observe [12]. Especially in the IBD area, it is used for differentiation of IBD from functional bowel disease, disease activity monitoring, prediction of therapeutic effect, prediction of recurrence, prognosis prediction, etc. [13,14,15] (Figure 1). The ideal biomarker is simple, non-invasive, fast, cost-effective, and reproducible [16]. Attempts have been made to accurately evaluate the condition of the intestinal tract of patients with IBD by blood and stool tests. However, it is also necessary to fully understand the usefulness and limitations of biomarkers [17].

C-reactive protein (CRP) and fecal calprotectin (FCP) have been reported as disease activity biomarkers for IBD [18]. CRP is widely used as a simple serum biomarker for predicting clinical activity in inflammatory diseases, including IBD. However, it is non-specific because of the elevation in systemic inflammatory diseases other than the intestinal tract [16,19]. FCP has been reported to be useful in various situations such as diagnosis of IBD, correlation with endoscopic severity, assessment of therapeutic effect, and prediction of relapse [19]. Nevertheless, in patients with UC, the correlation between FCP and clinical symptoms is inadequate at moderate to high disease activity [20]. The usefulness of FCP remains unclear because limited information is available in patients with CD [21,22,23]. In addition, the cutoff value is not standardized because of the change in cutoff value depending on the test method and kit [24]. The usefulness of the fecal immunochemical test (FIT) [25], which has been measurable for some time, and serum leucine-rich glycoprotein (LRG) [26] and urinary prostaglandin E-major urinary metabolite (PGE-MUM) [27], which have become measurable in patients with IBD in recent years, has also been reported. However, these are not well reported, and the evidence is poor. The erythrocyte sedimentation rate (ESR) rises slowly compared to CRP during inflammation, and ESR declines more slowly after improvements in inflammation [28]. The hypercoagulable state was associated with intestinal inflammation state, and serum fibrinogen level was correlated with the severity of the acute phase response [29]. However, in many studies, FCP and CRP are superior to fibrinogen and ESR for endoscopic remission in UC and CD [30]. Research results that combine multiple biomarkers and clinical symptoms have also been reported. In addition, conflicting results have been reported due to various clinical endpoints and cutoff values.

Here, we comprehensively review the usefulness and limitations of biomarkers that can be used in the daily clinical practice of IBD. This is the first review to describe the usage strategies including LRG and PGE-MUM that have recently become available in IBD daily clinical practice.

## 2. Treatment Goals for IBD

Initially, the main goal of medical treatment for IBD was to achieve stable clinical remission. However, MH is one of the major therapeutic targets specified in recent recommendations [31]. Mayo Endoscopic Subscore (MES) 0 is associated with superior disease outcomes while endoscopic healing in UC is commonly defined as MES ≤ 1. In CD, Turner et al. recommend that endoscopic healing in CD is defined as a Simple Endoscopic Score for Crohn’s disease (SES-CD) < three or absence of ulcerations while there is a lack of consistency in defining endoscopic remission [18]. Treat to Target (T2T) has been proposed as a therapeutic strategy aimed at improving the long-term prognosis of IBD. T2T is a strategy to set a nearby target and achieve it to improve the long-term prognosis in chronic diseases. Endoscopy is the current gold standard for monitoring MH in patients with IBD [32]. The recently updated Selecting Therapeutic Targets in Inflammatory Bowel Disease (STRIDE)-II states that the most important long-term achievable treatment goals for patients with IBD are clinical remission, endoscopic healing, quality of life recovery, and lack of disability [18]. Normalization of serum and fecal biomarkers are endorsed as a short-term goal. However, unlike hypertension and diabetes, where the concept of T2T was introduced earlier, frequent colonoscopy is difficult to evaluate the target due to invasiveness, human and medical economic restrictions, etc.

The search for superior biomarkers that are non-invasive, objective, and well-correlated with endoscopic disease activity is becoming more active in practicing the T2T strategy in IBD practice.

## 3. CRP

CRP is one of the proteins produced by hepatocytes during the acute phase reaction, mainly by stimulation with interleukin (IL)-6, and is a serum biomarker widely used in various inflammatory diseases. Due to its short half-life, it is widely used as an assessment of acute inflammation [33]. Although CRP is not disease-specific [34], it has long been used in IBD practice because it can be easily measured in blood in a short time [35]. A systematic review and meta-analysis by Menees et al. reported that the probability of having IBD in the normal range of CRP levels was less than 1% [36]. However, because CRP is a non-specific biomarker, it can be increased in conditions other than IBD, such as systemic infections and inflammatory diseases. Contrarily, some patients do not develop high CRP levels despite their active disease [37,38].

### 3.1. Evaluation of Disease Activity by CRP

CRP is more sensitive in CD compared to that in UC when assessing disease activity. In CD, the presence of active lesions is strongly suspected when CRP is positive due to its high specificity [39]. Moreover, Denis et al. reported that 92.9% of patients with CD with clinical symptoms had laboratory data that showed normal CRP, but the majority of their revealed lesions had mild inflammation. In this regard, they described that it is possible to rule out severe endoscopic lesions in patients with a clinically active CD if CRP is negative [40]. However, Henriksen et al. reported a poor association between disease phenotype and CRP in patients with CD [41]. Yoon et al. reported that CRP had a sensitivity range of 50.5–53.3% and a specificity range of 85.1–87.2%, and ESR had a sensitivity range of 68.7–71.3% and a specificity range of 63.4–66.4% for the detection of endoscopic remission using some endoscopic indices [42]. Endoscopic activity was better correlated with CRP than with ESR in patients with UC [42,43].

A review by Mosli et al. reported the ability to detect the endoscopic activity of IBD by CRP value, with a sensitivity of 0.49 (95% confidence interval [CI] 0.34–0.64) and a specificity of 0.92 (95% CI 0.72–0.96). Therefore, low CRP levels do not necessarily reflect that there is no endoscopic activity [24]. Ishida et al. evaluated the association between endoscopic scores of colonic inflammations and FCP, FIT, and CRP in patients with UC. FCP and CRP tended to correlate more strongly with the sum of Mayo Endoscopic Subscore (S-MES) and Ulcerative Colitis Colonoscopic Index of Severity (UCCIS) than with maximum Mayo Endoscopic Subscore (M-MES) and Ulcerative Colitis Endoscopic Index of Severity (UCEIS). In the M-MES ≤ 1, FC and FIT showed a strong correlation with S-MES and UCCIS compared to CRP. On the other hand, in the M-MES ≥ 2, only CRP was significantly correlated with S-MES and UCCIS [44].

### 3.2. Prediction of MH by CRP

CRP has not been able to provide sufficient accuracy to replace endoscopy as an independent biomarker for MH. Krzystek-Korpacka et al. investigated the role of CRP in the detection of MH in a review of 30 studies. CRP ranged from 0.4 to 28 mg/L, with large variations in the optimal cutoff value selected. The median sensitivity of CRP performance as an MH marker was 79.5% and the median specificity was 61% for CD, and the sensitivity was superior to the specificity. Regarding UC, the median sensitivity was 66% and the median specificity was 82%, and the specificity was superior to the sensitivity [45].

### 3.3. Prediction of Therapeutic Effect by CRP

#### 3.3.1. Prediction of Therapeutic Effect by CRP in CD

Louis et al. reported that in 226 patients with CD, anti-TNF-α agents were more effective in those with higher pretreatment CRP [46]. Reinisch et al. evaluated CRP levels at baseline and 14 weeks after infliximab (IFX) induction as predictors for maintained response or remission. CRP normalization 14 weeks after the induction of IFX increased the likelihood of maintained response or remission for 1 year [47].

Magro et al. reported that CRP levels 14 weeks after IFX induction in patients with CD were associated with a sustained response, independent of baseline CRP serum levels. However, unlike previous reports, high baseline CRP values correlated with worse responses [48]. The contradictory results are thought to be due to differences in CRP cutoff values and low albumin levels. A retrospective study of 1189 patients with CD by Tanaka et al. also found that high baseline CRP was associated with inadequate retention of adalimumab (ADA) treatment over a 4-year follow-up period [49].

#### 3.3.2. Prediction of Therapeutic Effect by CRP in UC

Reinisch et al. evaluated the remission rate of patients with moderate to severe active UC treated with ADA. In this multicenter, randomized, double-blind, placebo-controlled trial, a high baseline of high-sensitivity CRP (hsCRP) was associated with a reduced remission rate [50]. In a study of 72 patients with UC by Iwasa et al., improvement of clinical symptoms and reduction in CRP 2 weeks after IFX induction therapy were associated with subsequent prognosis [51]. 

Oxford criteria state that the risk of in-hospital colectomy is 85% if CRP exceeds 45 mg/L or if there are more than eight bowel movements in 24 h on the third day of intravenous corticosteroids [52]. Recent data have partially revisited these results. In-hospital colectomy rates decreased from 85% in 1996 to 36% in 2017 in a population of patients who met Oxford criteria [53]. The reason for this reduction in colectomy rates may be associated with the progress in the induction of remission for acute severe UC after failing with corticosteroid use.

### 3.4. Prediction of Recurrence by CRP

Regarding disease monitoring, there are several reports that CRP predicts clinical recurrence with CD. Consigny et al. measured CRP every 6 weeks in 71 patients with CD and reported that CRP predicted recurrence [54]. Bitton et al. reported that higher CRP was a predictor of relapse by measuring CRP every 3 months in patients with CD [55]. Roblin et al. conducted a prospective observational cohort study enrolling patients with IBD in clinical remission 14 weeks after the introduction of IFX therapy. It was reported that CRP > 5 mg/L 22 weeks after the introduction of IFX therapy predicted loss of response in patients with CD [56].

Conversely, contradictory data have been published regarding the correlation between CRP and postoperative recurrence in patients with CD. A study of 24 postoperative patients with CD showed no consistent association between endoscopic scores and CRP 54 weeks after surgery [57]. Another study indicated a weak but statistically significant difference in hsCRP between patients with postoperative recurrence and those with endoscopic remission within 18 months (median 7 months) of resection [58].

## 4. FCP

FCP is a calcium-binding protein consisting of a complex of two proteins, S100A8 and S100A9 [59]. FCP is primarily derived from neutrophils and has a direct antibacterial effect and a role in innate immune response [60]. FCP reflects the migration of granulocytes across the intestinal wall of patients with IBD [61]. FCP is stable at room temperature for several days [62] and its clinical utility as a biomarker has long been investigated [63]. FCP provides the strongest evidence as a single biomarker commonly used in the clinical practice of IBD [31,64].

### 4.1. Differential Diagnosis between IBD and Functional Bowel Disease

FCP is one of the sensitive non-invasive tools in distinguishing IBD from bowel dysfunction [64]. In a prospective multicenter study of colonoscopy in 870 consecutive patients, FCP was reported to have 89% sensitivity and 62% specificity in the diagnosis of organic disease. In patients referred for chronic diarrhea, the sensitivity and negative predictive value of FCP was 100% in the detection of any organic colonic disease [65]. van Rheenen et al. performed a meta-analysis to examine the accuracy of FCP for screening patients with suspected IBD. The pooled sensitivity and specificity of FCP were 0.93 (95% CI 0.85–0.97) and 0.96 (95% CI 0.79–0.99), respectively [66]. In a meta-analysis by Menees et al., there was a ≤ 1% probability of having IBD when CRP ≤ 0.5 mg/dL or FCP ≤ 40 μg/g [36].

### 4.2. Evaluation of Disease Activity by FCP

FCP values have been shown to correlate with endoscopic and histological activities in patients with IBD, and have emerged as a non-invasive tool for monitoring disease activity [20,67,68].

#### 4.2.1. Evaluation of Clinical Disease Activity by FCP

A meta-analysis by Lin et al. evaluated the diagnostic accuracy of FCP to distinguish between active and clinical remission in 744 patients with UC and 727 patients with CD in 13 studies. FCP cutoff values varied between studies and ranged from 30 μg/g to 274 μg/g. For a cutoff value of 50 μg/g, the pooled sensitivity and specificity were 0.92 (0.90–0.94) and 0.60 (0.52–0.67), respectively. For a cutoff value of 250 μg/g, the pooled sensitivity and specificity were 0.80 (0.76–0.84) and 0.82 (0.77–0.86), respectively [69].

Schoepfer et al. prospectively examined 228 patients with UC and 52 healthy controls and evaluated the correlation between endoscopic activity and FCP, CRP, hemoglobin, platelets, leukocytes, and clinical scores. The results showed that the endoscopic disease activity of UC was best correlated with FCP [70]. Schoepfer et al. evaluated the correlation between SES-CD and FCP, CRP, leukocytes, and CD activity index (CDAI) in patients with CD. FCP was most correlated with SES-CD [71]. In a study by Iwamoto et al., FCP was significantly correlated with inflammation of the intestinal tract as assessed by balloon-assisted endoscopy (BAE), even in patients with CD with only small bowel lesions [21]. However, FCP has also been reported to be less capable of assessing disease activity in patients with CD compared to patients with UC [69,72].

#### 4.2.2. Prediction of Endoscopic MH by FCP

There are many reports that FCP is also useful for predicting MH. In patients with UC, Kristensen et al. followed monthly FCP after baseline colonoscopy, up to two measurements of FCP < 250 μg/g or a follow-up of up to 12 months. Subsequently, flexible sigmoidoscopy was performed and MES was used to assess the degree of inflammation. All 16 patients who achieved FCP < 250 μg/g showed endoscopic MH (MES ≤ 1) [73]. Kawashima et al. reported that FCP showed a stronger correlation with S-MES than M-MES in patients with UC. S-MES is the sum of MES in five parts of the colon. Thus, FCP better reflects inflammation in the whole colon [74]. In patients with CD, Kawashima evaluated 70 BAEs in 53 patients. They showed that FCP was more accurate in predicting endoscopic remission of CD than hsCRP, albumin, white blood cell count, and platelet count. It was reported that the detection rate of MH was good with a sensitivity of 0.91 (95% CI 0.52–1.00) and a specificity of 0.82 (95% CI 0.52–098), even in patients with CD having only small intestinal lesions [22]. Monica, et al. conducted a systematic review to summarize published data on the performance of non-invasive biomarkers in assessing MH in patients with IBD. In UC, FCP cutoff levels for MH detection ranged from 58 mcg/g (sensitivity 89.7% and specificity 93.3%) to 490 mcg/g (sensitivity 100% and specificity 62%). In CD, FCP cutoff levels for MH detection ranged from 71 mcg/g (sensitivity 95.9% and specificity 52.3%) to 918 mcg/g (sensitivity 50% and specificity 100%). Although inferior in UC, FCP is also found to be well associated with MH in CD [75].

#### 4.2.3. Prediction of Histological MH by FCP

In recent years, treatment aimed at histological MH may be more effective in predicting patient outcomes than at endoscopic MH in patients with UC [76,77]. Previous studies have shown that FCP can be used to distinguish between patients with ongoing microinflammation and those with histological remission, in patients with UC who completely achieved endoscopic healing [20,78,79]. Magro et al. evaluated the Geboes score, Nancy Index, and Robarts Histopathology Index using biopsy samples from 377 patients with UC, and demonstrated that three histological classification systems could predict MES and FCP levels at sensitivity and specificity levels depending on the cutoff selected. In addition, higher FC levels were statistically associated with the presence of neutrophils in the epithelium and ulceration or erosion of the intestinal mucosa [80]. Recently, a systematic review involving 12 studies and 1168 patients confirmed the correlation between FCP and histological activity. However, it has been stated that identifying FCP cutoff levels requires larger prospective studies using validated histological indicators [81].

### 4.3. Prediction of Therapeutic Response by FCP

A large amount of data is accumulated to predict the therapeutic response by FCP.

In patients with remission UC with elevated levels of FCP, increasing the dose of mesalamine reduced FCP to those associated with lower rates of relapse [82]. De Vos et al. conducted a prospective study of 53 patients with UC undergoing the IFX induction cycle. Patients with significantly reduced FCP after 2 weeks were shown to be more likely to achieve endoscopic remission after 10 weeks [83]. Bertani conducted a prospective observational study of patients with UC who had begun biotherapy with IFX, ADA, golimumab, or vedolizumab. The FCP assessment 8 weeks after treatment with all biologics has been shown to be useful in predicting response to MH [84].

Baseline FCP levels may predict primary no response after induced IFX therapy in patients with CD [85]. In a report by Boschetti et al., consecutive 32 patients with CD were treated with IFX or ADA, and FCP levels at 14 weeks predicted clinical remission within 1 year after induction [86]. According to a report by Plevris et al., normalization of FCP within 12 months of diagnosis was associated with a reduced risk of disease progression in patients with CD [87]. Narula et al. enrolled a total of 677 patients with CD treated with ustekinumab, and showed the FCP after 6 weeks could predict endoscopic cure after 52 weeks, which may be more beneficial than the improvement of clinical symptoms [88].

However, FCP may not be useful for deciding therapeutic responce in a short period after starting induction therapy. Toyonaga et al. prospectively examined 31 patients with active UC. In their report, two-item patient-reported outcome, partial Mayo score, and Lichtiger clinical activity index were significantly reduced from day 3 of induction of remission in clinical responders after 4 weeks. However, FCP did not show a significant decrease until after 2 weeks [89].

### 4.4. Prediction of Recurrence by FCP

Among patients with IBD in clinical remission, patients with high FCP have been reported to be at higher risk of early relapse than those with low FCP [31,90,91]. Measurements of FCP over time also show that FCP is elevated prior to clinical relapse [92]. De Vos et al. measured the FCP of patients with UC receiving IFX maintenance therapy every 4 weeks. They reported that patients who experienced a flare had already significantly higher FCP levels 3 months before the flare-up [93]. Zhulina et al. prospectively examined 49 and 55 patients with CD and UC, respectively, and doubling FCP levels between two consecutively collected samples increased the risk of recurrence by 101% [94]. Molander et al. investigated whether elevated FCP levels after discontinuation of anti-TNF-α agents could predict clinical or endoscopic recurrence. They enrolled patients with IBD who were in remission after the start of anti-TNF-α agents. They followed up patients who discontinued anti-TNF-α agents and showed that FC levels were constantly elevated for a median of 94 (13–317) days before recurrence [92].

In a systematic review by Heida et al., asymptomatic patients with IBD who had repeated FCPs above the cutoff level had a 53–83% chance of recurrence within the next 2–3 months. Patients with consecutive normal FCP levels had a 67–94% probability to maintain in clinical remission in the next 2–3 months [95]. However, due to the limited number of studies that meet the selection criteria and the heterogeneity between the selected studies, the ideal FCP cutoff for monitoring could not be identified.

Wright et al. reported that FCP had sufficient sensitivity and negative predictive values to monitor CD recurrence after bowel resection [96]. Qiu et al. performed a meta-analysis of 613 patients with CD monitored by FCP after surgery. For predicting postoperative recurrence, the pooled sensitivity and specificity were 0.82 (95% CI 0.73–0.89) and 0.61 (95% CI 0.51–0.71), respectively [97].

### 4.5. How to Use FCP and Precautions

The final European recommendation for diagnostic assessment of IBD suggests FCP monitoring every 3 months if FCP is negative and monthly FCP monitoring if FCP exceeds the threshold [31]. If FCP increases in UC patients who maintain clinical remission using full dose biologic agents, we may need to perform endoscopy to consider treatment changes.

The problem of FCP is that the various cutoff value varies depending on the report. The reference values obtained from the results of clinical performance tests also differ depending on the test method and kit. Furthermore, it is known that the cutoff value differs depending on whether the aim is clinical remission, endoscopic remission, or histological remission. Therefore, the number of cutoff values used in each study is large [24,98,99]. In addition, there is a diurnal difference depending on the collection time even in the same case [100]. FCP increases with increasing defecation intervals; thus, it seems to be most appropriate to analyze the first stool in the morning [101]. In addition, intra-individual variability of FCP several days apart is significantly higher [102]. FCP has been reported to decrease in pregnant women with IBD [103]. FCP can be increased in many other intestinal conditions, including celiac disease, colon cancer, diverticulosis, intestinal infections, use of non-steroidal anti-inflammatory drugs, and proton pump inhibitors [66]. In addition, FCP also changes in several non-pathological conditions such as age, obesity, and lifestyle [104]. Obtaining fecal samples may be more difficult than obtaining blood or urine samples because some patients hesitate to bring stool samples to the hospital, and it is difficult for patients to collect stool samples from diarrheal stools. Therefore, patients usually prefer blood tests to fecal tests when given the option [105]. The disadvantage of FCP is that it is not available on the day of stool submission at many facilities.

## 5. FIT

FIT is a method for measuring stool hemoglobin concentration using an antibody specific to human hemoglobin. FIT is used worldwide as a screening method for colorectal cancer. It has been suggested that FIT may be useful in detecting mucosal lesions in IBD, especially UC. Sakata et al. reported that IBD was diagnosed in 35 of the 236,000 asymptomatic individuals who were FIT-positive in colorectal cancer screening [25]. However, there are few reports on its usefulness in IBD practice compared to FCP. FIT can be examined using smaller samples than those in FCP, and the measurement time of FIT is shorter than that of FCP. Moreover, the cost of FIT is cheaper than that of FCP [106]. In addition, the cut-off value of FIT is relatively stable compared to FCP, which is almost the same as the cut-off value used in screening for colorectal cancer [107].

### 5.1. Evaluation of Disease Activity by FIT

FIT correlates with endoscopic activity in UC and is particularly excellent in detecting MH [90,108]. Nakarai et al. evaluated the results of FIT in combination with 310 colonoscopies in 152 patients with UC. A large majority of patients with MES 0 were negative for FIT. When MES 0 was defined as MH, the diagnostic ability of FIT was 92% sensitive and 71% specific. When MES 0 or 1 was defined as MH, the sensitivity was 60% and the specificity was 87% [109]. Positive FIT (≥100 ng/mL) predicted mucosal inflammation (MES ≥ 2) with a sensitivity of 87% and specificity of 60%. Takashima et al. evaluated 105 colonoscopies in 92 patients with UC, along with FIT and FCP results. FCP and FIT had the same correlation as MES. FCP and FIT showed the same correlation as MES. When MH was defined as MES 0 or 1, the diagnostic capabilities of FCP and FIT were equivalent, but when MH was defined as MES 0 only, FIT was shown to be more sensitive [110].

Shi et al. reported that FIT was highly sensitive and accurate in predicting endoscopic and histological healing of UC [111]. Hiraoka et al. evaluated a total of 110 colonoscopies in patients with UC and examined the correlation between changes in colonoscopic findings and changes in FIT and FCP. FIT was more sensitive than FCP in predicting MH. Contrarily, FCP better reflects changes in endoscopic activity than FIT and is useful for monitoring mucosal status in patients with active inflammation [112]. 

Moreover, Ishida et al. reported that when the disease duration was ≥5 years, the correlation between FIT and endoscopic scores in patients with UC was weakened [113]. They speculated that intestinal fibrosis due to persistent chronic inflammation along with disease duration could affect the value of FIT in patients with UC because scarring tissue is less likely to cause bleeding.

FIT is not as useful in patients with CD as in patients with UC. Inokuchi et al. compared FIT, FCP, and SES-CS in 71 patients with CD (22 ileal type, 16 colonic type, 33 ileocolonictype). Both FIT and FCP were significantly correlated with SES-CD. However, in CD with the involvement of the small intestine, only FCP is significantly correlated with SES-CD, and FIT is less useful for evaluating the inflammation of the small intestine [114]. Ma et al. examined 40 patients with CD in a prospective cohort study undergoing colonoscopy and reported that the specificity of FIT that predicts MH for CD was relatively low [67].

### 5.2. Prediction of Recurrence by FIT

FIT is useful in predicting relapse of UC. Hiraoka et al. measured FIT at each hospital visit in 83 patients with UC in clinical remission and MH. No relapse was observed in 43 patients with persistent FIT negative (<100 ng/mL); however, relapse was observed in 25 patients (63%) among 40 patients with positive FIT (≧100 ng/mL) [115]. Nakarai et al. examined the clinical outcomes of 194 patients with UC in clinical remission who underwent colonoscopy. Negative FIT (≤100 ng/mL) results at least 1 year after induction of remission therapy were reported to correlate with complete MH (MES 0) and better prognosis [116].

### 5.3. How to Use FIT and Precautions

Despite inferior evidence compared to FCP, FIT seems to be more sensitive than FCP when it comes to assessing MH. However, FIT is inferior to FCP in monitoring the mucosal condition of patients with IBD having active inflammation. In addition, FIT becomes positive in intestinal bleeding other than IBD such as hemorrhoids.

Recurrence of UC localized to the right colon may reduce the usefulness of FIT, as the sensitivity to detect advanced tumors of the proximal colon is significantly lower than that of the distal colon [117].

## 6. LRG

LRG is a 50 kDa glycoprotein containing eight leucine-rich repeat domains, first reported by Haupt et al. in 1977 [118]. Serada et al. identified LRG by serum-based proteome analysis of patients with rheumatoid arthritis before and after treatment with anti-TNF-α agents [26]. It is expressed not only in hepatocytes but also in neutrophils, macrophages, and intestinal epithelial cells. Moreover, LRG is induced by inflammation caused by cytokines other than IL-6, such as TNF-α, IL-22, and IL-1β [119,120]. This contrasts with liver-derived CRP, which is produced in response to IL-6 released from macrophages and lymphocytes in the intestinal tract [121]. Yoshimura et al. also reported increased LRG gene expression in peripheral blood mononuclear cells in UC and CD, indicating that it reflects clinical disease activity [122]. It is expected to reflect intestinal inflammation more directly compared to serum CRP because LRG is produced from cytokine-stimulated neutrophils and epithelium associated with inflammation of the intestinal tract of CD and UC. LRG can be increased in many other inflammatory diseases: psoriasis [123,124], gastric cancer [125], colorectal cancer [126], heart failure [127], diabetes [128], obesity [129], and prognosis in primary biliary cholangitis [130]. More studies are required for the clinical usefulness of LRG in IBD.

### 6.1. Evaluation of Disease Activity by LRG

It has been demonstrated that serum LRG levels were correlated with disease activity in patients with CD and UC and that LRG was also elevated in patients with active CD and UC with normal CRP [26,119]. Shinzaki et al. showed that LRG was useful in detecting endoscopic MH in 129 patients with UC. LRG levels in patients with UC experiencing histological healing were also shown to be significantly lower than those in patients without histological healing [131].

Serada et al. also showed that serum LRG correlated with the CDAI in 22 patients with CD. In addition, some patients with CD with clinical or endoscopic activity but with normal serum CRP were positive for serum LRG [26]. Yoshimura et al. prospectively enrolled 98 and 96 patients with UC and CD, respectively. Serum LRG levels were positively correlated with clinical disease activity and CRP levels in patients with UC and CD. However, in CD, the correlation between LRG and SES-CD did not reach statistical significance. One of the causes of this discrepancy is believed to be that SES-CD, including evaluation of only colon and observable ileum, was inadequate to assess the overall small bowel lesion activity [122].

Sinzaki et al. conducted a prospective study (PLANET study) to investigate the usefulness of LRG as a monitoring biomarker for IBD. Serum LRG levels decreased with improved clinical and endoscopic outcomes during ADA therapy [132]. In a PLANET sub-study, Yanai et al. showed that LRG could be a marker for predicting ADA trough levels in ADA-treated patients with CD and UC. Patients were enrolled at the time of starting ADA therapy and evaluated for 52 weeks or until drug withdrawal. ADA trough levels at week 12 or ADA withdrawal were negatively correlated with LRG at the start of ADA treatment, which was statistically significant. CRP or FCP was not associated with ADA trough levels [133].

LRG is expected to be a biomarker for IBD, but controversial results have also been reported. A recent study by Kourkoulis et al. showed that no significant differences were observed in LRG levels between patients with UC and healthy controls [134]. Yoshimura et al. further investigated the diagnostic accuracy for detecting endoscopic remission in UC. Although the Mayo score values for defining endoscopic remission were different (MES 0 or MES ≤ 1), AUC was similar in LRG and CRP, unlike previous results showing that AUC of LRG was higher than the AUC of CRP [122]. Yasutomi et al. evaluated LRG performance with respect to the predictability of MH in consecutive 166 patients with UC and 56 patients with CD and compared it with those of CRP, FIT, and FCP. the performance of LRG was not superior to that of FIT and FCP. Conversely, in patients with CD, the performance of LRG was equivalent to that of CRP and FCP. The reason for the difference from the previous study was a large number of enrolled patients with low disease activity. Fecal markers could be preferable for patients with IBD with low disease activity compared to LRG [135].

### 6.2. Effects of Anti-TNF-α Agents on LRG Levels

It is controversial whether LRG is a suitable biomarker to follow up patients with anti-TNF-α agents. Anti-TNF-α agents may affect serum LRG levels, as TNF-α is one of the cytokines that promote LRG production. Yoshimura et al. reported that serum LRG was lower in patients receiving anti-TNF-α agents than those in patients not receiving it [122]. The study by Sinzaki et al. suggests that LRG may help monitor disease activity even after anti-TNF-α agents [132]. Yasutomi et al. reported that in patients with UC, the area under the curve (AUC) of LRG in patients who received anti-TNF-α agents was not superior to that of patients who did not receive it. In patients with CD, the AUC of LRG in patients who received anti-TNF-α agents was higher than that in patients who did not receive it [135].

## 7. PGE-MUM

PGE-MUM is a new biomarker that reflects the endoscopic activity of UC. Prostaglandin E2 is produced at the site of inflammation and released into the blood, but its half-life in blood is short, and thus, accurate measurement is difficult. Therefore, PGE-MUM, which is a highly stable urinary metabolite, attracted attention [136,137].

PEG-MUM levels can be elevated in smokers, patients with chronic lung disease, and patients with cancer [138,139,140,141]. Laxatives have also been reported to increase PGE-MUM levels [136]. Due to its non-invasiveness and simplicity, it is expected that evidence will be accumulated in the future.

### 7.1. Evaluation of Recurrence Predictability by PGE-MUM

Arai et al. evaluated whether PGE-MUM could be used as a biomarker of UC activity. UC activity was assessed by the simple clinical colitis activity index of the 99 patients and the MES and Matts’ grading of 79 patients. Both PGE-MUM and CRP levels were correlated with UC activity. PGE-MUM was a significant independent predictor of histological remission, but CRP was not [27]. PGE-MUM also reflects the endoscopic score of pediatric patients [142]. Ishida et al. reported that PGE-MUM is useful as a biomarker that reflects endoscopic activity comparable to FIT, particularly in patients with UC with long-term disease duration. PGE-MUM and FIT were measured on 92 urine and fecal samples from 60 patients with UC. The FIT was significantly correlated with MES in a disease duration <5 years, but not in a disease duration ≥5 years. Conversely, PGE-MUM was significantly correlated with MES in both a disease durations. PGE-MUM shows a stronger correlation with S-MES than with MES [143]. Ishida et al. conducted a prospective observational study to assess the relationship between changes in endoscopic scores and changes in PGE-MUM and CRP levels in patients with UC. A significant association between the change in endoscopic scores and PGE-MUM levels was observed. However, there was no corresponding increase in CRP with higher endoscopic scores [144].

### 7.2. Prediction of Recurrence by PGE-MUM

Ishida et al. reported whether PGE-MUM could predict clinical recurrence of UC. Sixteen of the 70 patients relapsed during a 12-month follow-up. The median PGE-MUM value at admission in patients with UC with recurrence was significantly higher than that in patients with UC in clinical remission, particularly, in patients with UC with long-term duration. This study also showed that the recurrence rate predicted by PEG-MUM values was more accurate than that of the endoscopic score [145].

## 8. Combination of Biomarkers

Due to the complexity and heterogeneity of IBD, a single marker cannot predict disease activity in all cases. Therefore, the possibility of combining different biomarkers and clinical scores to improve accuracy is being investigated.

### 8.1. Evaluation of Disease Activity by Combined Biomarkers

Minderhoud et al. reported that the Utrecht Activity Index, which combines daily liquid stool frequency with CRP, FCP, platelet count, and mean platelet volume, provided optimal results for predicting endoscopic activity in patients with CD [146]. For values of FCP 100–250 µg/g, which are difficult to interpret in clinical settings, the integration of FCP with CRP and clinical data may help classify patients with uncertain disease activity [147]. Post-hoc analysis of the CALM study reported the performance of the combination of FCP, CRP, and CDAI in the detection of MH [148].

NLR has been extensively reported in systemic inflammation and malignant condi-tions. Assessment of NLR is non-invasive, low cost, and is easily calculated from blood count data routinely. The higher NLR values were associated with higher clinical disease activity, in patients with both CD and UC [149]. However, in assessing endoscopic activity, NLRs were not as useful in CD patients as in UC patients [150,151,152].

### 8.2. Prediction of Therapeutic Effect by Combined Biomarkers

Sollelis et al. reported that a decrease in FCP, CRP, and clinical remission after 12 weeks were the predictors of corticosteroid-free remission at 52 weeks in a cohort of patients with CD treated with anti-TNF-α agents [153]. Choy et al. reported that the CRP/albumin ratio after IFX salvage therapy predicts therapeutic response. The ratio also can predict colectomy among patients with acute severe UC [154]. The CALM trial is an open-label, randomized, and controlled Phase 3 trial conducted in 74 hospitals and outpatient centers in 22 countries. Patients with CD treated with ADA were divided into two groups and prospectively followed. The MH rate after 1 year was significantly higher in the group in which treatment was strengthened based on the T2T strategy using FCP and serum CRP as monitoring tools in addition to clinical symptoms than in the group in which treatment was strengthened based only on clinical symptoms [155]. Dulai et al. examined the role of FCP in monitoring clinical and endoscopic responses in patients with UC treated with biologic agents or tofacitinib induction cycles (6–8 weeks). It has been stated that endoscopy may be avoided if rectal bleeding is resolved, stool frequency is normalized, and FCP ≤ 50 μg/g [156].

## 9. Novel Biomarkers

In recent years, many data on biomarkers that are expected to be used in the daily clinical practice of IBD have accumulated [157,158]. MicroRNAs (miRNAs) are a single-stranded RNA of about 21 to 25 bases that negatively regulates gene expression [159]. Several miRNAs in the sample of patients with active IBD that differ from those in patients with inactive IBD and controls [160]. Glycosylation is a common and complex posttranslational modification of proteins. Total plasma N-glycomes expression patterns were reported to be associated with disease features and the need for treatment [161]. Cytokine oncostatin M (OSM) belongs to IL-6 subfamily. Several types of immune and stromal cells can produce OSM. It was reported that high levels of OSM in the mucosa were strongly associated with the severity in patients with IBD [162]. B-cell activating factor (BAFF) is a type II transmembrane protein belonging to the TNF family. BAFF is mainly expressed in innate immune cells such as monocytes, dendritic cells, macrophages, and neutrophils [163]. Responders to IFX treatment had higher BAFF levels at baseline compared to non-responders [164].

## 10. Use of Biomarkers Properly

The gold standard for understanding the disease activity of IBD remains endoscopy. However, biomarkers will be more useful in daily clinical practice by understanding the characteristics of each biomarker and clarifying the target and purpose because they are less invasive and simpler than endoscopy. In addition, we also need to consider costs. The cost of FCP and LRG are cheaper than endoscopy, but more expensive than CRP and FIT (Table 1).

## 11. Limitation of Biomarkers

Ideal biomarkers should be non-invasive, sensitive, disease-specific, easy to implement, and cost effective [16]. However, there are currently no biomarkers that can replace endoscopy to meet these requirements. To date, most biomarkers other than CRP and FC lack validation in large populations. Biomarkers also target a wide variety of study designs and heterogeneous patient groups. The same biomarker has different observation periods, and definitions of clinical remission, MH, and recurrence. Therefore, further studies with a larger number of patients will be required to establish the cut-off level of each new biomarker.

## 12. Conclusions

With the progress of IBD treatment, the use of biomarkers is being applied not only to diagnosis and monitoring but also to personalized treatment. Currently, no biomarker is an alternative to endoscopy. However, the use of appropriate biomarkers avoids frequent endoscopy, reduces the psychological and physical burden on the patient, and leads to cost savings. We strongly expect the emergence of new biomarkers for patients with IBD in the future. Therefore, it is important to understand the characteristics of each biomarker and use the optimal biomarker at the right timing in daily clinical practice.

## Figures and Tables

**Figure 1 life-11-01375-f001:**
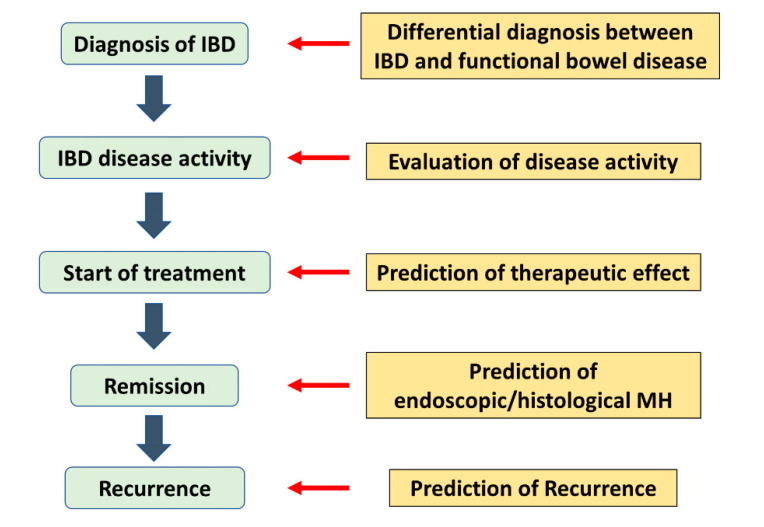
The role of biomarkers in the practice of IBD. Biomarkers are used in a variety of situations, from diagnosis to treatment of inflammatory bowel disease. IBD: inflammatory bowel disease, MH: mucosal healing.

**Table 1 life-11-01375-t001:** Characteristics of biomarkers and endoscopy in inflammatory bowel disease practice.

Biomarker	Sample	Invasion	Expensive	Strong Point	Weak Point	Evidence
CRP	blood	+	+	Can be measured repeatedly in a short period Excellent evaluation of activity in the acute phase	Low sensitivity (especially UC) Non-specific rise	+++
FCP	stool	-	++	Excellent judgment of MH Can be positive in the active phase with negative CRP	Large error in the acute phase Not suitable for the evaluation of advanced inflammation Non-specifically elevated in the blood The cutoff value is not fixed Evidence is slightly poor in CD	+++
FIT	stool	-	+	Excellent judgment of MH Can be positive in the active phase with negative CRP	Large error in the acute phase Not suitable for the evaluation of advanced inflammation Non-specifically elevated in the blood	++
LRG	blood	+	++	Higher correlation with disease activity than CRP Can be positive in the active phase with negative CRP	Evidence is slightly poor in CD May not be suitable for cases with low disease activity	+
PGE-MUM	urine	-	No data	Less invasive and convenient	Poor evidence in CD	+
Endoscopy		+++	+++	Gold standard for monitoring MH	High invasion and costs	+++

CD: Crohn’s disease; CRP: C-reactive protein; FCP: fecal calprotectin; FIT: fecal immunochemical test; LRG: leucine-rich glycoprotein; MH: mucosal healing; PGE-MUM: prostaglandin E-major urinary metabolite; UC: ulcerative colitis; -: none; +: low degree; ++: moderate degree; +++: high degree.

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
