# Peer review of "Role of Biomarkers in the Diagnosis and Treatment of Inflammatory Bowel Disease"

_life, 2021, doi:10.3390/life11121375_

Round 1

Reviewer 1 Report

Congratulations. Excellent review. Well-written, comprehensive and educational. I just have a few observations.

  1. Line 121: Please spell out MES for the first time (you spell out MES on Line 122).
  2. Line 420: Please remove a space between “biomarker” and “for IBD” (there are two spaces).

Reviewer 2 Report

  • “However, it is non-specific because of the elevation in systemic inflammatory diseases other than the intestinal tract”

There are several non-pathological conditions that can lead to altered FC values, too (cite “Caviglia GP et al. Fecal calprotectin: beyond intestinal organic diseases. Panminerva Med. 2018 Mar;60(1):29-34. doi: 10.23736/S0031-0808.18.03405-5. Epub 2018 Jan 25. PMID: 29370679.”)

  • Use oxford comma in the whole text

  • Definite MH in CD

  • Compare the sensibility of CRP and ESR in UC

  • What is your advice in a patient with UC in clinical remission with a full dose biologic (already done dose escalation) with increasing values of calprotectin?

  • What about the costs of the different biomarkers?

  • What about fibrinogen? Neutrophils to lymphocytes ratio?

Reviewer 3 Report

In this study, authors reviewed the role of available biomarkers in the management of inflammatory bowel disease. Review is clearly written, with good flow of information. However, a few comments should be addressed before potential publication:

My main comment is associated with CRP and FCP sections – as they are already well-established biomarkers, used in a daily clinical practice, and have been investigated robustly over the years, in my opinion, these sections should be written in somewhat more concise way, with more limited sub-sections. Better point, for the future, are other collected markers, and future perspectives of biomarker use (that could be expanded more in the last part of the review).

Furthermore, authors could make a distinction between CRP and high-sensitivity CRP in this review, as numerous studies use hs-CRP.

As this review is collecting information regarding new biomarkers that could be of use in IBD diagnosis and management, authors should add an additional short section regarding B Cell-Activating Factor (BAFF), that is emerging to be useful as well.

Round 2

Reviewer 3 Report

The authors have been responsive, and quality of the manuscript improved. 

I have no further comments.